# Defining the Role of Metastasis-Initiating Cells in Promoting Carcinogenesis in Ovarian Cancer

**DOI:** 10.3390/biology12121492

**Published:** 2023-12-05

**Authors:** Ji Wang, James C. Ford, Anirban K. Mitra

**Affiliations:** 1Indiana University School of Medicine-Bloomington, Indiana University, Bloomington, IN 47405, USA; jwa7@iu.edu (J.W.); jimmyford523@gmail.com (J.C.F.); 2Melvin and Bren Simon Comprehensive Cancer Center, Indiana University, Indianapolis, IN 46202, USA; 3Medical and Molecular Genetics, Indiana University School of Medicine, Indianapolis, IN 46202, USA

**Keywords:** ovarian cancer, metastasis, metastasis-initiating cells, microenvironment

## Abstract

**Simple Summary:**

Metastasis is the main cause of poor outcomes in ovarian cancer and metastasis-initiating cells are a subpopulation of cancer cells that can effectively establish metastases. We provide an overview of their characteristics, function, and the potential of targeting them in ovarian cancer.

**Abstract:**

Ovarian cancer is the deadliest gynecological malignancy with a high prevalence of transcoelomic metastasis. Metastasis is a multi-step process and only a small percentage of cancer cells, metastasis-initiating cells (MICs), have the capacity to finally establish metastatic lesions. These MICs maintain a certain level of stemness that allows them to differentiate into other cell types with distinct transcriptomic profiles and swiftly adapt to external stresses. Furthermore, they can coordinate with the microenvironment, through reciprocal interactions, to invade and establish metastases. Therefore, identifying, characterizing, and targeting MICs is a promising strategy to counter the spread of ovarian cancer. In this review, we provided an overview of OC MICs in the context of characterization, identification through cell surface markers, and their interactions with the metastatic niche to promote metastatic colonization.

## 1. Introduction

Ovarian cancer (OC) represents the most lethal gynecologic malignancy, characterized by a bleak prognosis [1,2]. Most patients are diagnosed at an advanced stage in which peritoneal metastases and the concurrent development of ascites contribute to OC-related mortality. While most patients respond well to the standard therapeutic regimen involving tumor debulking surgery and multiple cycles of carbo-taxol chemotherapy [1,3], recurrences are common, eventually leading to the development of chemoresistance.

Unlike many other cancer types relying on hematogenous routes for metastasis, transcoelomic metastasis is the predominant mode of spread in OC patients [4]. Metastasizing to the greater omentum is very common in OC and requires precise coordination between OC cells and the metastatic microenvironment [4,5]. Initially, cancer cells exfoliate from the primary tumor, develop resistance to anoikis, develop the ability to attach to and invade into the omentum, and subsequently reprogram their metabolism toward the use of lipids abundantly available at the metastatic site. Changes in the OC cells include the elevation of CPT1A, sustained activation of ERK signaling pathways, the continued high expression of Bcl-X_L_, the losses of some cellular connections, and the adoption of a more mesenchymal phenotype through epithelial-mesenchymal transition (EMT) [6,7,8,9,10,11]. Simultaneously, fluids are accumulated inside the peritoneal cavity to form ascites, facilitating the mobility of detached OC cells, and providing them with more pro-metastatic signals [12]. These OC cells subsequently aggregate to form multicellular spheroids floating in the ascites, escaping immune surveillance, and avoiding chemotherapeutic drugs [13]. Based on the abdominal anatomical features and tropisms of OC cells towards adipose tissues, spheroids are more likely to colonize in the greater omentum [14]. The adhesion of OC cells to the mesothelial layer is initiated by interactions among integrins, surface proteins on OC cells (CA125, CD44, CD133 and CD24), and mesothelin [15,16,17,18,19,20,21,22]. Throughout this process, OC cells enhance the secretion of MMP-2, promoting the degradation of vitronectin and fibronectin. As MMP-2 cleaves these proteins, resulting fragments induce the heightened binding capacities of α5β1 and αVβ3 integrins to the cell surfaces, thereby elevating the migration and invasion potential of OC cells [23,24]. In addition, the downregulation of E-cadherin in metastatic OC cells induces the expression of α5-integrin, facilitating the binding of α5β1 integrin to cleaved fibronectins at the metastatic site. Fibronectin is highly expressed by mesothelial cells covering the omentum and the fibroblasts in the underlying basement membrane. Consequently, the activated α5β1 integrin induces c-Met in a ligand independent manner causing Src and FAK phosphorylation, promoting invasion through the mesothelium and the basement membrane of the omentum [25,26]. After successfully invading the mesothelium, OC cells commence establishing clinically significant metastases by coordinating with extracellular matrices (ECMs), immune cells, and stromal cells [27,28,29].

Detached OC cells usually undergo a long, risky, and complicated journey of metastasis. Previous research on clonal evolution has revealed that only a small percentage of exfoliated OC cells possess the capability to thrive at metastatic sites [30]. Those cells harboring the potential to give rise to metastasis are called metastasis-initiating cells (MICs), which are poorly understood. Unraveling the mysteries surrounding the MICs of OC will contribute to the development of new therapeutic interventions to enhance the quality of life in OC patients.

In this review, we commence with an overview of the heterogeneity observed in metastatic OC cells. We then describe the characteristics of MICs and potential markers to identify them. In addition, we summarize the effects of the metastatic niche on MICs, and potential therapeutic strategies against them.

## 2. Heterogeneity of Metastasis

OC encompasses a heterogeneous spectrum of diseases [31]. Despite the classification of OC into three main subtypes based on origin (epithelial OC, sex-cord stromal OC, and germ cell OC), variations within the same subtype, particularly in high-grade serous OC (HGSOC), are readily apparent. The Cancer Genome Atlas (TCGA) has categorized HGSOC into 4 transcriptomic subtypes with significantly different clinical outcomes based on bulk mRNA sequencing data from 489 primary samples: immunoreactive, differentiated, mesenchymal, and proliferative [32]. Subtyping based on bulk sequencing implies the variability of tumor cells and their surrounding microenvironment. Comparative bulk genomic analyses between primary HGSOC samples and corresponding metastases demonstrated that genetic heterogeneity and intra-tumor heterogeneity increase in metastases, emphasizing the intricate signaling dynamics within the metastatic microenvironment and the need for further research into heterogeneity of metastasis [33].

The introduction of single-cell sequencing techniques has ushered in a new era of research, shifting from a low-resolution tissue-based sequencing analysis to a high-resolution cell-based investigation. It enables scientists to investigate the characteristics of different cell subclusters and differentiate malignant cells from normal cells in the TME (stromal and immune cells) within a single sample [34,35,36]. A study led by Siel Olbrecht et al. has demonstrated that there are many cell subclusters with distinctive transcriptomic profiles in metastasis, including one subcluster marked by the high expression level of BAMBI (closely correlated with tumor growth, tumor invasion and carboplatin resistance) along with the upregulation of EMT and IL2/STAT5 signaling pathways. Notably, two critical subclusters upregulating KRT6A and MAGEC2, respectively, influencing multiple signaling pathways, such as the p53 pathway, adipogenesis, glycolysis, DNA repair, oxidative phosphorylation, the WNT/β-catenin signaling pathway, and the TGF-β signaling pathway [37]. Research led by Dr. Aviv Regev has also identified a variety of different cancer cell clusters, in which inflammatory cytokine modules (represented by IL6, IL8 and TNF), cell cycle modules, and MHC class II modules are differentially enriched [38]. In addition, both teams found that most subclusters are patient-specific, which may be partly attributed to various baseline expression levels from different patients [31,32]. Beyond the variability within malignant cells in metastasis, the diversities of stromal and immune cells are also important to the heterogeneity of OC [39]. Multiple fibroblast subclusters were identified and differentially specialized in inflammatory responses, extracellular matrix productions, and other important biological patterns [40,41].

Constant changes of the tumor microenvironment result in the emergence of cell subclusters with various genetic and transcriptomic profiles, and it undoubtedly bolsters the metastatic milieu’s capacity to adapt to its surroundings, prompting a fundamental question—where do they come from?

## 3. Metastasis-Initiating Cells

About 120 years ago, the seed and soil hypothesis, formulated by Dr. Stephen Paget, has pointed out the organ-specific secondary growth of cancer cells in breast cancer, instigating extensive research on the factors leading to metastasis in different types of cancer [42]. Two decades later, a hierarchical hematopoietic model was proposed by Dr. Maximow [43], ushering in a new era of stem cell research. Hematopoietic stem cells (HSCs) were identified, and their regenerative characteristic was confirmed by Dr. Becker, Dr. Till, and Dr. McCulloch, indirectly leading to the isolation of AML (acute lymphoid leukemia)-initiating cells and preliminary characterization of them in 1994 [44]. Several decades later, the self-renewal and differentiation features of AML-initiating cells were validated by Dr. Dick, providing the bedrock for the contemporary cancer stem cell (CSC) hierarchical model [45]. Building upon the concept of stemness, Dr. Bapat successfully isolated cancer initiating cells from the ascites of a patient diagnosed with grade IV ovarian serous adenocarcinoma in 2005 [46]. Shortly thereafter, transcoelomic metastasis, a special cancerous dissemination route, was clearly described as a multi-step process for ovarian cancer by Dr. Tan [4]. This delineation spurred subsequent research, placing a heightened focus on understanding the initiation of metastasis in OC. Hosseini et al. established a breast cancer mouse model to recapitulate early lesions of cancer metastasis through expressing the Her2 transgene at puberty of BALB-NeuT mice [47]. MICs in OC, a concept in its relative infancy, chiefly encompasses a group of cells endowed with the capacity to establish clinically significant intraperitoneal metastasis.

### 3.1. Origins of MICs

Extensive research has provided compelling evidence for the existence of a specific type of cancer cells, including those in OC, capable of metastasizing to other parts of the body from primary tumors [14,48,49]. Nevertheless, the precise origins of these metastasis-capable cells remain a subject of ongoing debate. At the forefront of this debate is the quest to determine the primary tumor’s origin. Histologically speaking, HGSOC cells bear a resemblance to normal cells lining the fallopian tube (the most anatomically proximal structure), endometrium, and endocervix, leading researchers to hypothesize that OC cells originate from fallopian tube epithelia (FTE). However, some researchers insist that OC cells originate from ovarian surface epithelia (OSE) because of proximity and their epithelial phenotype [50]. Detailed profiling of OSE-derived and FTE-derived OC cells revealed that the latter exhibits a more aggressive phenotype coupled with heightened chemoresistance and a proclivity for invasion [51]. MICs are more likely to originate from FTE-derived OC cells, but whether MICs intrinsically attain their initiation capacity in the beginning or acquire it later remains unknown and more research needs to be done.

### 3.2. Characteristics of MICs

To thrive in the dynamic microenvironments and to establish the distant omental metastasis, MICs must possess not only the potential to colonize remote sites but also the remarkable ability to swiftly adapt to external stressors. The defining characteristics of MICs can be comprehensively examined through three distinct perspectives: markers, multi-omic profiles, and the intricacies of their metastatic niches.

#### 3.2.1. Markers

The isolation of MICs from tumor masses is a pivotal step for researchers to characterize them and to find potential therapeutics targeting them, relying heavily on the selection of markers. Those markers can be categorized into surface markers and intracellular markers. Surface markers are particularly favored over other cellular markers due to their ease of recognition by specialized antibodies without compromising the integrity of the OC cell membranes. This approach facilitates the sorting of a larger population of live cells through flow cytometry techniques, enabling more comprehensive characterizations. Sometimes, some intracellular markers (enzymes and transcription factors) can also be utilized to identify MICs, for they can offer valuable insights into the functional profiles of these cells. In the forthcoming section, we will review the markers employed for identifying MICs.

##### CD24

CD24 is a sialoglycoprotein, originally found to be widely expressed on the surface of B cells to prevent them from terminal differentiating into plasma cells [52]. Because of its intrinsic trait to modulate differentiation, CD24 is also a well-known ovarian cancer surface marker, predicting a poor prognosis in OC patients. High expression of it is found to be positively correlated with an advanced FIGO stage and presence of peritoneal metastasis [53]. In addition, CD24 is a marker for OC CSCs [54,55,56], capable of initiating cancerous lesions in the peritoneal cavity, confirming the metastasis-initiating trait of CD24+ cells. In terms of cancer-initiating capacity, CD24 can increase the phosphorylation of STAT3 via JAK2 to favor the growth of primary tumor spheres and the ability of forming secondary spheres [57], playing an important role in forming spheroids in ascites during metastatic dissemination. Resistance to anoikis (the first step of transcoelomic metastasis) is also closely related to the high expression level of CD24 [4]. In addition, adhesions to the surface of mesothelial cells, a necessary step for invading the peritoneum, are mediated by interactions between CD24 and P-selectins, a mesothelial surface protein [58,59].

##### CD44

CD44, a glycoprotein ligand, plays a pivotal role in mediating cell-to-cell adhesion through its interactions with a range of molecules, including hyaluronic acids (HAs) [20,60,61], osteopontin [61,62], collagens [63,64] and metalloproteinases (MMPs) [65,66]. There are multiple isoforms of CD44 that are confirmed to be correlated with cancerous phenotypes in various cancers [67,68,69]. Notably, research has highlighted the metastatic initiation potential of CD44 variant 6 (CD44v6) in advanced ovarian cancer (OC) [48,70] with negative correlations observed between good prognoses and the enrichment of CD44v6 positive OC cells [20,71,72]. Furthermore, the mesenchymal isoform of CD44 is able to promote EMT of OC cells and endow them with a stem cell-like phenotype [73]. During the adhesive step of the transcoelomic metastasis of OC, a sandwich model of two CD44 receptors and one stabilized HA/Versican polymer in between them has been proposed to emphasize the supportive role of CD44 during the invasion of the peritoneum [20]. Interestingly, CD44 was found to be transferred from one subpopulation of OC to the others to augment the metastatic potential in low-metastatic cells via exosome-mediated transport [74]. A combination of both CD44 and CD117 has been employed to isolate cells with heightened stemness and increased metastatic capacity [48].

##### CD133

CD133 is a glycoprotein originally found to be expressed on adult stem cells and widely known to play a critical role in maintenance of stemness [75,76,77]. That stemness-associated characteristic is also applied to OC and previous research has suggested the metastatic-initiation role of CD133+ cells [78]. Furthermore, CD133 is closely associated with EMT mediated by ZEB2 and negatively correlates with good prognoses in OC patients in terms of overall survival (OS) [79]. CD133 can also contribute to the metastasis of OC cells via increasing secretion of MMPs and upregulating the surface markers (PECAM1 and ICAM1) to strengthen adhesions [80]. It can also trigger the transcriptional activity of β-catenin to maintain CSC properties via interacting with integrins in various cancer types. Whether CD133+ cells preserve the CD133-β-catenin pathway remains an intriguing area for exploration in OC [81].

##### CD117

CD117 (c-KIT) is a receptor tyrosine kinase protein and it is normally upregulated in mast cells and certain hematopoietic stem cells, which is closely related to cell differentiation [82,83,84]. Studies have demonstrated that CD117+ OC cells possess the capability not only for self-regeneration but also for generating diverse metastatic variants [85]. Multiple reports have established direct correlations between CD117 expression and various factors such as chemoresistance, advanced clinical stage, malignancy, as well as markers like SOX2, COT4, and NANOG [86,87,88]. Furthermore, the higher expression level of CD117 in extracellular vesicles from OC cells is positively correlated to more aggressive tumor invasion [89]. Given that CD44 also serves as an initiation-associated factor, the combination of CD117 and CD44 has been applied to identify cell subpopulations with relatively higher stemness potential [90,91,92].

##### ROR1

ROR1 is a receptor tyrosine kinase-like orphan receptor, primarily recognized for its involvement in the modulation of neurite growth, exhibiting higher expression levels in embryonic tissues compared to adult counterparts [93,94,95,96,97]. Some previous reports have indicated that high expression levels of ROR1 have detrimental effects on the prognoses of OC patients in terms of overall survival and progression-free survival rates [98]. Kipps et al. linked stemness to ROR1 and found that more spheroids and a higher expression level of ALDH1 are positively correlated with the enrichment of ROR1+ OC cells [99]. Moreover, the inhibitory effects of silencing ROR1 on invasion and adhesion of peritoneal membranes are validated in vitro, further implicating the potential metastatic initiation capability of ROR1+ cells [100]. Nevertheless, research employing in vivo models and metastatic experiments is warranted to comprehensively elucidate the functions of ROR1+ cells in OC.

##### ALDH

The family of aldehyde dehydrogenases (ALDH) comprises numerous isoforms responsible for catalyzing the oxidation of aldehydes [101,102]. ALDH1A1, notably identified as a negative prognostic indicator, is a well-established marker for OC CSCs [103,104,105,106] and previous research has shown that ALDH1A1 contributes to the maintenance of stemness via the coordinated regulation of cell cycle checkpoints and DNA repair networks [107]. It has also been observed that ALDH+ cells are enriched in patient ascites-derived spheroids, implicating the resistance to anoikis imparted by ALDH. In addition, those spheroids finally contribute to the formation of metastatic lesions in the peritoneal cavity [108]. The inhibition of conversion from ALDH- OC cells to ALDH+ ones, mediated by DDB2, can decrease peritoneal metastasis, suggesting the metastasis-initiating capability of ALDH [109].

##### SOX2

The Yamanaka factors, consisting of SOX2, KLF4, OCT4, and MYC, have been pivotal in mediating the induction of pluripotent stem cells from terminally differentiated fibroblasts—an epochal achievement in stem cell research that has reverberated into the study of cancer stem cells [110,111,112,113]. OCT4 and SOX2 were identified to be core transcription factors, whereas KLF4 and MYC to be essential factors in maintaining pluripotency [110]. Most importantly, only a small percentage of cells were confirmed to co-upregulate the expression levels of all 4 of the above factors [114], coinciding with the widely observed phenomenon that cancer-initiating cells account for less than 10% of all cells within the tumor mass. In the context of ovarian cancer (OC), SOX2 assumes a predominant role in the maintenance of cancer stemness, particularly within spheroids, as evidenced by its higher expression levels [88,115]. Notably, SOX2 exhibits a dual role in OC, enhancing metastatic potential while diminishing adhesive capabilities [116], implicating its important role in metastasis initiation. While numerous publications have linked the malignancy of OC to elevated MYC and OCT4 expression levels [117,118,119,120,121], none of them have explicitly indicated the metastasis-initiating capacity of both factors. Interestingly, KLF4 plays a tumor suppressing role during metastasis of OC and previous research has also explicitly indicated its apoptosis-inducing effects and chemo-sensitizing functions [122,123]. Even though the transcriptional activity of SOX2 (the most effective Yamanaka factor for MIC markers) can be applied to identify some MICs, one main limitation of utilizing TFs as markers is that they cannot be used to isolate live cells by FACS.

#### 3.2.2. Multi-Omic Profiles

In the realm of cancer research, the widespread utilization of genomic, epigenomic, and transcriptomic techniques has empowered the creation of cellular profiles with unprecedented resolution. We will review the potential of comprehensively characterizing MICs by harnessing the capabilities offered by these three aspects—genomic, epigenomic, and transcriptomic.

##### Genomic Profiles

Many studies have implicated the enrichment of metastatic-initiating cells with stemness in spheroids in OC and suggested that CD44+/CD117+ cells possess the potential to initiate metastasis. Comparative genome analysis has been done to compare the CD44+/CD117+ cells with CD44−/CD117− cells isolated from the ascites of six patients and found that there were small genetic differences (rearrangements on chromosome 2) between them in terms of single nucleotide polymorphisms (SNPs) in just one patient [124], indicating potential differentiation from double positive cells to negative cells. Interestingly, hundreds of somatic genomic arrangements between primary and metastatic tumor samples were identified and several somatic breakpoints influenced the cancer-related genes (FANCD2, ERBB4 and ESR1) [125], substantiating genomic instability and variability during metastasis, but under most circumstances, global genomic instability is rarely observed, implicating the selection for MICs (accounting for a small percentage of the whole primary tumor) under external stresses. Recent extensive research involving 250 biopsy pairs from a diverse range of metastatic solid malignancies, including OC, has explored genomic changes in metastases under treatment selection pressure from a pan-cancer perspective. Interestingly, the genomic profiles of metastases remained nearly unchanged during treatment in terms of known actionable genomic points. However, substantial variations emerged in the genomic profiles of metastases during treatment, primarily in the unactionable genomic regions, especially intergenic regions., implicating the influences of epigenomic differences induced by those changes [126]. In summary, relative genomic stability is the major characteristic of MICs regardless of whether they are under treatment or not, and trivial changes of genomic profiles of MICs and non-MICs also confirms that those non-MICs may potentially originate from MICs.

##### Epigenomic Landscape

Epigenetic regulations of gene expression are key to cellular phenotypical changes [127,128,129,130], and they are more flexible and immediate, compared to genomic changes. Histone modifications [131,132] (such as histone methylation, histone acetylation, and histone phosphorylation), DNA modifications [133,134] (such as DNA methylation and DNA hydroxyl methylation), and non-coding RNA-induced transcriptional changes [135,136,137] mainly account for epigenetic regulatory pathways. During normal differentiation, the global openness of the chromatins of stem cells is decreased and the transcriptional activities of most pluripotent genes will be diminished [138]. Similar to normal adult stem cells, the stemness of MICs is related to global hypomethylation and local hypermethylation of specific genes and that trait imparts great flexibility of manipulating transcriptions of genes to MICs in response to changing external stimuli [139]. In MICs, the variance of the expression levels of all genes is expected to be lower than that of non-MICs and entropy is used to quantitatively assess the stemness of cells [140,141,142]. The higher the entropy value (the quantitative measure for uncertainty) is, the more the number of possible differentiation lines that cells can walk down. Multiple methods have been developed to quantify entropy values. Given that the phenotypes of cancer cells rely heavily on the activation/inhibition of different combinations of various signaling pathways, some researchers calculate the signal entropy values based on the protein–protein interaction (PPI) network and gene expression profiles [143]. Additionally, Dr. Vaidya has developed a method to calculate entropy values based on DNA methylation, where the variation in these values reflects the replication status of stem cells [144].

In addition, PRC2 (polycomb repressive complex 2) is a very important regulator in the epigenetic field and its main job is to close target genomic regions via methylating histone H3 on lysine 27 [145,146,147]. MICs can effectively use EZH2, the histone methyltransferase of PRC2, to mediate the decrease of DAB2IP, a negative regulator of WNT signaling via causing the downregulation of WNT5B, to maintain their stemness. MICs can also leverage DNA methylase and HOTAIR, a non-coding RNA which associates with PRC2 to play a suppressive role in transcription, to achieve the same goal [148].

Previous research on OC also suggests that global histone acetylation and hypomethylation can augment the stemness of OC cells to generate more MICs [149]. Furthermore, histone lysine demethylases are responsible for upregulating SOX2 to elevate the initiation potentials of MICs via modifying the chromatin landscape in OC [150].

##### Transcriptomic Landscape

The transcriptomic characteristics of MICs can be summarized into two categories: plasticity and stemness. Transcriptomic plasticity is mainly imparted by flexible epigenomic regulations, and it can endow cells with the ability to switch between EMT and MET (mesenchymal epithelial transition) [151,152,153,154]. EMT, a well-studied biological process, is required for cancer cells to acquire a mesenchymal phenotype to invade into stromal tissues via breaking the mesothelial layer and pave the way for metastatic colonization [155]. There are multiple signaling pathways (Wnt/β-catenin signaling pathway, PI3K/Akt/mTOR signaling pathway, TGF β signaling pathway) which can activate EMT [156,157,158]. The Wnt/β-catenin signaling pathway includes canonical and non-canonical pathways, the former of which is studied heavily and requires the destruction of the β-catenin complex to mediate the transcription of downstream genes [157]. Cysteine-rich intestinal protein 1 (CRIP1) and BRMS1 = like transcriptional repressor (BRMS1L) have been identified to be associated with EMT via augmenting or diminishing the expression of Wnt/β-catenin-mediated genes (TCF, MYC and CCND1) in OC cancer cells, respectively [159,160]. In addition, previous research has found that PCNP can promote the EMT of OC cells via increasing the accumulation of β-catenin in the nucleus. Interestingly, the simultaneous acquisition of stemness during the EMT process has been reported in many cancer types including OC, implicating the importance of EMT in MICs [161]. In contrast to EMT, MET can restore OC cells back to an epithelial phenotype with more proliferative potentials, which is necessary for cancer cells to spread throughout the greater omentum, but there is a very limited number of publications about MET in OC cancer cells. Previous research has found that SMAD7 is a negative regulator of the TGF-β signaling pathway and the SMAD7-mediated-MET phenomenon was observed in OC stem cells [162]. Importantly, the process of transitioning between epithelial and mesenchymal phenotypes is not dichotomous but rather comprises a spectrum of intermediate cellular states known as hybrid EMT cells [161,163]. However, whether MICs are hybrid EMT cells remains to be explored.

The stemness trait lies at the heart of cancer-initiating cells, engaging MICs in the potential differentiation lines towards other cell types with different functions, and the lower proliferative capacity of this trait makes MICs less susceptible to standard chemotherapeutic drugs. Except for Yamanaka factors, epigenetic regulators should be implementors of this program, flexibly and constantly switching chromatic regions on or off to upregulate or downregulate specific downstream genes [109,164,165,166,167,168]. Previous research has found that KDM4C can upregulate OCT4 expression to increase the stemness of OC cells via the trimethylation of lysine 9 of histone 3 at the promoter region of OCT4 [167]. In addition, DDB2, responsible for the trimethylation of lysine 27 of histone 3, was discovered to bind to the ALDH1A1 gene promoter to interfere with the dedifferentiation process in OC, implicating its role in the differentiation of MICs [109].

#### 3.2.3. Metastatic Niche

Successful metastatic initiation hinges on the intricate coordination of multiple cell types to build a microenvironment (niche) suitable for MICs to proliferate, differentiate, and maintain (Figure 1). However, the composition and architectural features of these niches can vary significantly depending on the target organ and cancer type. The greater omentum, which consists of a great number of adipocytes, is the most preferable metastatic site for OC cells, where it is lined by a double layer of mesothelial cells, a first-line defense for cancerous invasion [4,169]. In addition, milky spots (specialized secondary lymphoid organs), aggregates of immune cells (T cells, B cells, granulocytes, and macrophages), stromal cells (fibroblasts, the high endothelium of veins, and mesothelial cells), and extracellular matrices (ECMs), are widely distributed across the greater omentum. Under normal circumstances, milky spots mediate immunity against foreign bodies in the peritoneal cavity whereas they often initiate insufficient anti-tumor immunity and finally become accomplices of metastatic cancer cells to augment the progression of metastatic initiation [170,171]. Furthermore, tumors are classified into hot, variable, and cold based on the infiltration of immune cells and their pro-/anti-inflammatory traits, informing therapeutic targeting of immune checkpoints [172]. OC tumors enriched in pro-inflammatory immune cells manifest better prognoses compared to ones having regulatory T cells, which can also be used for predicting responses to immunotherapies [173,174]. The formation of the immune landscape of OC is a dynamic process mediated by the elimination and escape phases of immunoediting. Throughout the process, cytotoxic functions, elicited by CD8+ T cells, NK T cells, NK cells, and IFN production, facilitate the annihilation of tumor cells with recognizable antigens. However, the failure of the induction of killing them finally leads to the survival of tumor cells which upregulate multiple ligands to evade normal cancer immunosurveillance [175]. In this section, we delve into the mechanisms through which different cell types within the metastatic niche collaborate to enhance the metastatic initiation potential.

##### Fibroblasts

Cancer-associated fibroblasts (CAFs) are a heterogeneous group of abnormal fibroblasts, which are capable of assisting metastatic cancer cells in invading normal tissues [176,177,178]. Cancer-associated fibroblasts (CAFs) constitute a diverse group of abnormal fibroblasts within the tumor microenvironment, and their origin—whether they arise from the reprogramming of normal fibroblasts, pre-existing cell populations within normal fibroblasts, or other mesenchymal stem cells—remains an area of active investigation [179]. The main job of CAFs is the secretion of factors that can further facilitate the migration, invasion, initiation potentials, and proliferation of MICs. Epidermal growth factors (EGF) secreted by fibroblasts can boost the size of spheroids by upregulating the α5 integrin expression on the surface of MICs enabling the recruitment and adhesion of these cancer cells to the floating cell masses [180]. Moreover, activated CAFs can also secrete POSTN under the induction of TGF-β1 to potentially mediate the EMT in MICs via PI3K/AKT pathways [181], implicating the progression of metastasis. Previous research has also found that CAFs can augment the FGF4 expression and initiate potentials in MICs via paracrine FGF4-FGFR2 signaling [182].

##### Mesothelial Cells

Mesothelial cells serve as the first defense against the metastasis of OC, lining the peritoneal cavity. They are easily assimilated by OC cells to become accomplices in metastatic invasion (cancer-associated mesothelial cells), contracting themselves and exposing the sub-mesothelial tissues under the influence of microenvironmental factors [183,184,185,186,187]. Mesothelial-to-mesenchymal transition, a main metastasis-precipitating program, is initiated in the mesothelial cells contacted by OC cells and CAF-activating factors, such as alpha-SMA and FAP, are expressed at higher rates in mesothelial cells than before, simultaneously making the peritoneal membrane less slippery and more adhesive [188]. Interestingly, Tadashi Kimura et al. shows that CD44, one of the MIC markers, can be transferred to mesothelial cells via exosomes to mediate the increasing expression level of MMP-9, responsible for the degradation of extracellular matrices, and the decreasing expression level of E-cadherin [189]. Reciprocally, mesothelial cells are also able to change the transcriptomic profile of OC cells to adapt more quickly to metastasis and previous research has indicated that stemness can be augmented by OC spheroids cocultured with mesothelial cells [190]. Moreover, osteopontin, an important factor secreted by cancer-associated mesothelial cells, was confirmed to increase the intrinsic stemness phenotype in the cells via activation of CD44 receptors and the PI3K/AKT signaling pathway [191], implicating the potential to generate more MICs.

##### Adipocytes

Adipocytes are the most prevalent cells in the greater omentum, playing an important role in the establishment of omental metastasis in OC. They are also potential associates of OC cells, fueling OC cells with a copious quantity of lipid nutrients to meet the needs of high proliferative capacities [192], where previous research has suggested that rigorous fatty acid oxidations can direct OC cells towards a more proliferative phenotype with increased survival capability [192,193]. Intriguingly, lipid desaturases, which convert saturated fatty acids (usually enclosed inside the lipid droplets absorbed by OC cells) to unsaturated ones, are identified to be a key metabolic factor to increase the stemness of cells via increasing the transcriptional activities of the NF-kB signaling pathway, suggesting the strong link between adipocytes and MICs [194]. Even though there are many publications revolving around the interactions between metastatic OC cells and adipocytes, there is nearly no other research studying the direct communication between MICs and adipocytes, and more remains to be explored.

##### Macrophages

Macrophages are categorized into 2 main types (M1 and M2) based on their functions towards tumorigenesis in OC [195,196,197]. Macrophages with an M2-like phenotype are more likely to induce an immune-suppressive microenvironment to favor the colonization of OC metastasis whereas those with an M1-like phenotype maintain the normal immunity against tumor cells to increase survival rate in advanced OC patients [198,199]. Polarization of macrophages towards M1 (higher M1/M2 ratio) is confirmed to be correlated with increased glycolytic activities and decreased heme synthesis, and the latter suggests the limited utility of irons by OC cells [200]. Previous research also emphasizes the pro-tumorigenic role played by M2 macrophage cells [196,198]. F4/80+/CD206+ M2 macrophages are not only part of the ascitic spheroids but also augment the formation of spheroids, suggesting their pro-metastatic functions on MICs inside the spheroids since cancer initiating cells are also enriched in them. Proliferation-associated markers are identified to be upregulated by them, implicating that M2 macrophages can mediate the differentiation of MICs towards proliferative phenotypes. In addition, those M2 macrophages are found to be coincidentally enriched in the center of the spheroids charactered by an abundance of quiescent cancer cells, increasing the possibility of contact between MICs and macrophages [201]. In omental metastasis, macrophages with M2-like characteristics can also increase the activities of the stemness phenotype and EMT of MICs [202]. Reciprocally, UBR5 (ubiquitin protein ligase E3 component), overexpressed in OC cells, has been confirmed to sustain the β-catenin signaling pathway, an important upregulator of stemness in MICs that mediates the infiltration of M2 macrophages, emphasizing the link between them [203].

##### T Cells

In ovarian cancer (OC), tumor-infiltrating T cells encompass several distinct types, each characterized by specific surface markers, including CD4+, CD8+, proliferative, and unconventional T cells [204]. Under normal circumstances, CD4+ and CD8+ T cells should be responsible for initiating anti-tumorigenic immunity, but they are exhausted by the surrounding tumor microenvironment via upregulating PD-1 receptors [205,206]. The binding of PD-L1 and PD-L2 to PD-1 receptors leads to the impairment of normal TCR signaling pathways and corresponding co-stimulation signals. In addition, TIGIT and Tim-3 receptors have been identified to surface on the membranes of tumor-infiltrating T cells to mediate immune-suppressive functions [206]. Interestingly, the in vitro induction of T cells can have killing effects on OC cells with high stemness [207], implicating the cytotoxic effects on MICs and corroborating the opinion that T cells are modulated to become less aggressive by OC cells. Moreover, IL-10, an anti-inflammatory factor secreted by regulatory T cells, is determined to be increased in coculture with MICs compared to monoculture and CCL5-CCR5 interactions are found to mediate that process. Regulatory T cells are also induced to secrete more MMP-9 to degrade extracellular matrices, laying a good foundation for subsequent establishment of metastases [208]. Notably, regulatory T cells in OC exhibit an enhanced immune-suppressive capacity due to the elevated expression levels of specific markers, such as PD-1, 4-1BB, and ICOS, when compared to other cancer types [209].

##### B Cells

B cells are usually classified based on their developmental lineage [210,211,212] (early pro-B cell, late pro-B cell, large pro-B cell, large pre-B cell, small pre-B cell, and immature B cell), functional status [211] (plasmablast, B-1 cell, regulatory B cell, B-2 cell, plasma cell, and memory B cell) or localization [213] (follicular B cell and marginal zone B cell). During OC metastasis, the infiltration rate of B cells capable of secreting tumor-specific IgG is positively correlated with good prognoses via anti-tumor immunity, implicating the potential promotional effects on the production of cytotoxic T cells where they, most of which are memory-like B cells, are found in tertiary lymphoid structures (TLSs), surrounded by T cells and dendritic cells [214]. Resembling lymph nodes, those B cells, located in the center of the TLS, are expectedly proliferative. Interestingly, plasma cells can skew OC cells towards a mesenchymal phenotype via secreting some exosomes containing micro-330-3p, which mediates the junctional adhesions among cells, implicating the EMT modulation on MICs [215]. In addition, Biragyn et al. found that regulatory B cells (known to initiate immuno-suppressive effects favored by MICs) are surprisingly differentiated by precursor B cells that immigrate from bone marrow to cancerous sites, and that the intriguing phenomenon is mediated by thymic stromal lymphopoietin (abnormally secreted by OC cells) to increase the mobility of precursor B cells [216]. However, there is no research explicitly focusing on the direct interactions between B cells and MICs, and it can only be speculated that B cells act like a double-edged sword for MICs in OC depending on the types of B cells involved.

##### Neutrophils

Neutrophils are a type of granulocytes and neutrophil extracellular traps (NETs, originally thought to limit the mobility of microbes), mobilized and extruded by stimulated neutrophils, play a controversial role in the regulation of metastasis of OC cells. Previous research has indicated that NET formation is closely related to improved prognoses in OC, whereas other research shows that NETs augment the omental metastasis of OC cells via citrullination of histones mediated by PAD4 enzymatic activity [217,218,219]. It has been observed that NETs can act like a magnet, attracting OC cells to adhere to it where they are enriched in the premetastatic omental niche, implicating that it can also potentially assist MICs in colonization of omental tissues [218]. In addition, neutrophils can accelerate EMT to make OC cells more mesenchymal with a preference towards clustering together via secreting elastase to degrade E-cadherins [220], suggesting potential effects on MICs. However, there is no publication directly revolving around the interactions between MICs and neutrophils in OC, which may be a new direction for future research on MICs in OC.

### 3.3. Therapeutic Strategies

This part of the review will focus on targeting MIC treatment in four (4) therapeutic strategies: immunotherapy, anti-EMT therapy, epigenetic therapy, anti-stromal therapy, and marker-targeted therapy.

#### 3.3.1. Immunotherapy

Coordination with the immune-suppressive microenvironment to establish metastasis is very common in various cancer types [221,222,223,224] and there are many therapeutics aimed at restoring the normal functions of the immune system. PD-1/PD-L1 targeted therapy has been applied in OC treatment, which can effectively remove the inhibitory effects on normal anti-tumor immune functions. Single agent (Nivolumab, Avelumab and Pembrolizumab) use is confirmed to be effective against OC to some extent [225,226,227] but their efficacy is limited. The efficacy of PD-1/PD-L1 therapy can be enhanced by the addition of other drugs (Cisplatin, Bevacizumab, Doxorubicin, Olaparib) [228,229,230,231]. In addition, adoptive T-cell therapy (ACT) is another new field in the immunotherapy of cancer [205,232,233] even though it has not been widely used in OC. The main advantage of ACT is that antigen-specific T cells (generated by ex vivo techniques) are exempt from the inhibitory effects of the tumor suppressive immune-microenvironment, and there are multiple clinical trials utilizing this technique [205,234,235]. If MICs are further characterized, antigen-specific T cells against MICs can be selected to improve the prognoses of OC patients.

#### 3.3.2. Epigenetic Therapy

Epigenetic therapy has been widely used in many cancer types [236,237,238]. The flexibility of epigenetic regulations has endowed MICs with the potential to differentiate to other cell types. Targeting important factors involved in this differentiation process is very promising in improving the prognosis of OC patients. Therapies targeting DNMTs (DNA methyltransferase) and histone acetylation have gained clinical benefits in OC [239,240]. A clinical trial (NCT03348631) utilizing Tazemetostat (an EZH2 inhibitor) is still ongoing for OC treatment.

#### 3.3.3. Targeted Therapy

Some surface markers that have been identified on MICs are able to be treated utilizing two main types of marker-targeted therapy: transportation-based and function-based. Transportation-based therapies rely on the markers to locate the cells expressing corresponding markers and transfer drugs to eliminate them. In one study, Dr. Hu engineered nanomicelles with CD133 aptamers to transfer Gefitinib to lung cancer stem cells, which has not been applied in OC treatment [241]. Nevertheless, it is applicable to engineer nanomicelles with aptamers targeting one or more markers of MICs to accurately identify them. In addition, nanoparticles targeting CD44 have been successfully invented to direct drugs to eliminate target cells [242] and it will be of great clinical significance for future OC treatments. Function-based therapies can change the transcriptomic profiles of cells via inhibiting or elevating specific pathways. Metformin was confirmed to eliminate some MICs via changing methylation landscapes in an OC clinical trial [243]. Moreover, various inhibitors targeting ALDH have been applied to ovarian cancer experimentally [107,109,244], but surprisingly, none of them have been widely used in OC treatment. We speculate that the reason that these inhibitors are not widely used is that it is too difficult to differentiate normal cells from OC cells, increasing potential toxicity during treatment.

#### 3.3.4. Anti-EMT Therapy

EMT is a complex process involving many signaling pathways [245]. The TGF-β and Wnt-catenin signaling pathways are two of the prominent regulators. Therapeutically targeting both signaling pathways can consequently inhibit EMT from occurring in MICs or other malignant cells. In terms of the TGF-β signaling pathway, anti-TGF-β R I (Vactosertib, Galunisertib), anti-TGF-β R II (Bintrafusp alfa, LY3022859), and TGF-β (Fresolimumab) therapies have been applied in various clinical trials [246,247,248,249,250], but they are not widely used in OC currently. For the Wnt-catenin signaling pathway, ipafricept, which targets Wnt ligands to downregulate Wnt-catenin signaling pathway, was applied in the phase I clinical trial of OC and has shown to limit the progression of OC [251]. Many other signaling pathways are also involved in EMT and therapeutically targeting them can also potentially contribute to improved prognosis in OC patients. Furthermore, anti-EMT therapy can also have potential inhibitory effects on stromal cells (mesothelial cells, fibroblasts, and adipocytes), regardless of their association with cancer progression.

## 4. Conclusions

In summary, MICs in OC exhibit several critical characteristics, including the ability to survive various challenges during the metastatic journey, a level of stemness that enables differentiation into different cell populations, and the capacity to interact with the microenvironment to establish supportive niches (Figure 2). While there is substantial research on ovarian cancer stem cells, the specific study of MICs is limited. Nonetheless, there are common features and markers shared between MICs and OCSCs, raising the question of whether MICs should be considered a subgroup of OCSCs. Currently, only a single study has identified MICs at the single-cell level using sequencing techniques, with limited samples and no experimental validation in OC. For future studies, a combination of single cell sequencing techniques and cell sorting techniques will be required to unveil more details about MICs and identify more actionable targets to help prevent metastasis from occurring or progressing in OC. Targeting MICs in therapeutic strategies could inhibit OC from successfully metastasizing, effectively improving patient outcomes. However, many treatment options have yet to be tested for their effectiveness against OC, and a significant portion of current therapies primarily target OC after MICs have already established themselves in the greater omentum.

## Figures and Tables

**Figure 1 biology-12-01492-f001:**
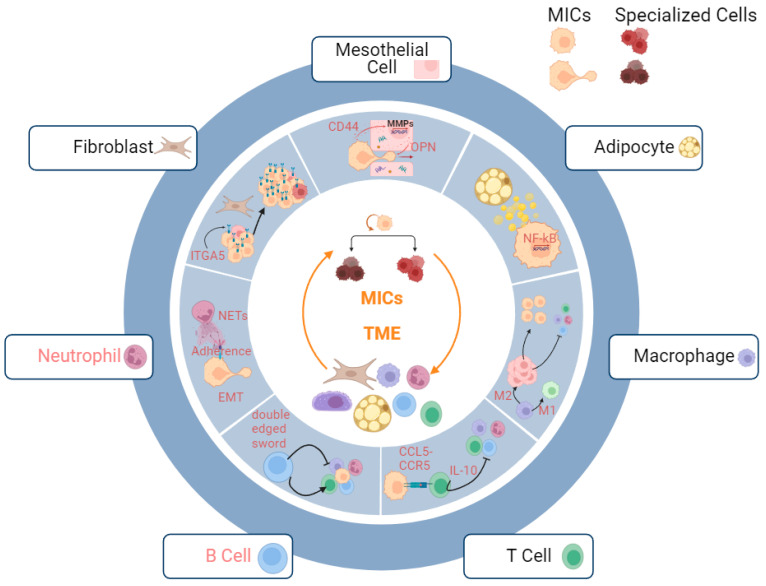
Metastatic niche of MICs. Productive interactions with multiple stromal and immune cells help MICs in establishing metastases. Mesothelial cells: OPN secreted by mesothelial cells facilitate the invasion of MICs; adipocytes: lipid droplets secreted by adipocyte amplify the stemness of MICs via upregulating NF-kB signaling pathways; macrophages: M2 macrophages assist immune escape of MICs by mediating suppression of pro-inflammatory immune microenvironment; T Cells: binding of CCR5 and CCL5 suppresses pro-inflammatory immune microenvironment; B Cells: B cells serve as both pro-inflammatory and anti-inflammatory mediators for cancer immune microenvironment; neutrophils: neutrophils increase adhesions of MICs to omentum via NETs; fibroblasts: fibroblasts facilitate the adaptation of MICs to microenvironment via upregulating ITGA5 and provide a supportive niche. The interactions between B cells/ neutrophils (red highlight) and the MICs remain to be explored.

**Figure 2 biology-12-01492-f002:**
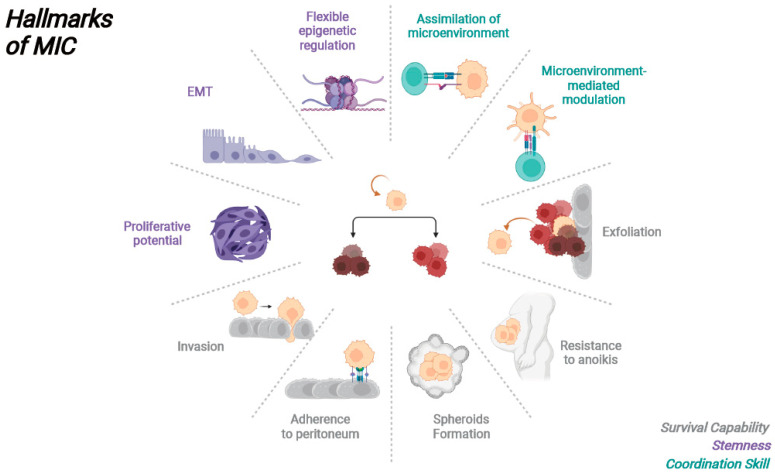
Hallmarks of MICs: schematic overview of the key characteristics of MICs that enable them to establish metastases at the distant sites.

## Data Availability

Not applicable.

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
