# Peer review of "Defining the Role of Metastasis-Initiating Cells in Promoting Carcinogenesis in Ovarian Cancer"

_biology, 2023, doi:10.3390/biology12121492_

Round 1

Reviewer 1 Report

Comments and Suggestions for Authors

Wang et al. provide a comprehensive review on the topic of metastasis initiating cells in ovarian cancer, cells which have some stem-like features.  Much of the review focuses on ovarian cancer cells undergoing metastasis and what features are necessary for this process. This includes not only the cancer cell, but also the microenvironment. The authors finish the review by discussing some possible future treatments. There are only minor changes suggested to make the review more accessible. 

CD117 is brought up as being important and a marker, but not described as to what it is.  It would be helpful to include some info on what it is/does why that plus CD44 may be stem cell markers in OC.

Possibly give more info about entropy measurements in epigenetics section

All Figure legends should be more descriptive to explain the models, enough so you can understand what you are looking at.  In figure 1, it’s unclear what are the cancer cells versus the other cells and if they are showing interaction with these other cell types.  I do not think that figure 2 is adding much to the text and should be modified or removed. If modified, perhaps those 4 types could be connected in the figure with the different features of MICs 

Line 396, EGF is epidermal growth factor, not epithelial growth factor. If showing EGF as an example, that should be clarified.  Also, the rest of that sentence is unclear. Isn’t EGF upregulating ITGA5 on cancer cells not recruiting it? Please update this to better reflect that paper cited. 

T cells – HGSOC is often thought to be a cold tumor.  Are there any differences in metastasis with “hot” and cold tumors or increased T-regs? Might be helpful to discuss that a bit more to clarify the differences.

No acknowledgement section was included

Comments on the Quality of English Language

Only some minor issues, such as addition of s to words that do not need it.  

Author Response

CD117 is brought up as being important and a marker, but not described as to what it is.  It would be helpful to include some info on what it is/does why that plus CD44 may be stem cell markers in OC.

We have now included a paragraph on this in the manuscript (Lines 227-237)

Possibly give more info about entropy measurements in epigenetics section

Entropy measurement methods have now been described in lines 336-342.

All Figure legends should be more descriptive to explain the models, enough so you can understand what you are looking at.  In figure 1, it’s unclear what are the cancer cells versus the other cells and if they are showing interaction with these other cell types.  I do not think that figure 2 is adding much to the text and should be modified or removed. If modified, perhaps those 4 types could be connected in the figure with the different features of MICs 

We have addressed these issues in the revised figure legends. As suggested, we have removed Figure 2 the revised manuscript has only 2 figures.

Line 396, EGF is epidermal growth factor, not epithelial growth factor. If showing EGF as an example, that should be clarified.  Also, the rest of that sentence is unclear. Isn’t EGF upregulating ITGA5 on cancer cells not recruiting it? Please update this to better reflect that paper cited. 

This has been addressed now in lines 432-435.

T cells – HGSOC is often thought to be a cold tumor.  Are there any differences in metastasis with “hot” and cold tumors or increased T-regs? Might be helpful to discuss that a bit more to clarify the differences.

This has now been discussed in lines 410-420.

No acknowledgement section was included

We have added an acknowledgement section.

Comments on the Quality of English Language

Only some minor issues, such as addition of s to words that do not need it.  

We have edited the manuscript for typos.

Reviewer 2 Report

Comments and Suggestions for Authors

Within this context the authors should also describe in detail the potential role of the basal membrane, the extracellular matrix and tumor micronenvironment. Furthermore, regarding metastatic dissemination following model should also be taken into account: Hosseini H, et al. Early dissemination seeds metastasis in breast cancer. Nature. 2016;540(7634):552-558.  

Author Response

Within this context the authors should also describe in detail the potential role of the basal membrane, the extracellular matrix and tumor micronenvironment.

This has been added to the manuscript (lines 54-64).

Furthermore, regarding metastatic dissemination following model should also be taken into account: Hosseini H, et al. Early dissemination seeds metastasis in breast cancer. Nature. 2016;540(7634):552-558.  

We have added this in lines 138-140.

Reviewer 3 Report

Comments and Suggestions for Authors

In this review article, Wang J, et al have summarized information about the pathophysiological significance of metastasis-initiating cells in context of ovarian cancer disease progression. They have provided a very detailed description about the markers that are employed to identify this sub-population of ovarian tumor cells. The authors have also addressed the complex tumor microenvironment encountered by these cells for establishment of metastases & have included detailed discussion of therapeutic strategies to target this cell type. Overall, the review is quite extensive and includes the most up to date information on this topic.

Comments on the Quality of English Language

The review article is well written. Minor language modifications may be required.

Author Response

Overall, the review is quite extensive and includes the most up to date information on this topic.

We really appreciate the positive review and have edited the manuscript for typos.

Round 2

Reviewer 2 Report

Comments and Suggestions for Authors

The comments have been addressed satisfactory.